# Singlet Oxygen Generation Driven by Sulfide Ligand Exchange on Porphyrin–Gold Nanoparticle Conjugates

**DOI:** 10.3390/ijms24087600

**Published:** 2023-04-20

**Authors:** Akira Shinohara, Hideyuki Shinmori

**Affiliations:** 1Polymer Chemistry Group, Sagami Chemical Research Institute, Yokohama 252-1193, Japan; shinohara.akira@nims.go.jp; 2Department of Biotechnology, Faculty of Life and Environmental Science, Graduate Faculty of Interdisciplinary Research, University of Yamanashi, Yamanashi 400-8510, Japan

**Keywords:** singlet oxygen (^1^O_2_), photosensitizer, ligand exchange reaction, gold nanoparticle, porphyrin

## Abstract

Here, we report a switching method of singlet oxygen (^1^O_2_) generation based on the adsorption/desorption of porphyrins to gold nanoparticles driven by sulfide (thiol or disulfide) compounds. The generation of ^1^O_2_ by photosensitization is effectively suppressed by the gold nanoparticles and can be restored by a sulfide ligand exchange reaction. The on/off ratio of ^1^O_2_ quantum yield (Φ_Δ_) reached 7.4. By examining various incoming sulfide compounds, it was found that the ligand exchange reaction on the gold nanoparticle surface could be thermodynamically or kinetically controlled. The remaining gold nanoparticles in the system still suppress the generation of ^1^O_2_, which can be precipitated out simultaneously with porphyrin desorption by the proper polarity choice of the incoming sulfide to restore the ^1^O_2_ generation.

## 1. Introduction

Singlet oxygen (^1^O_2_), a kind of reactive oxygen species (ROS), has been employed for a myriad of applications due to its mild reactivity, long lifetime and unique electron spin properties. Among the methods for generating ^1^O_2_, a photosensitizing reaction is used in many research fields [1,2,3,4,5,6,7]. This reaction is based on energy transfer from an excited triplet state photosensitizer molecule to a ground state triplet molecular oxygen (^3^O_2_), in which porphyrins are representative and highly efficient photosensitizers. Switchable photosensitization triggered by specific external stimuli is a potential requirement [8,9,10,11,12], as other photo-processes such as fluorescence, phosphorescence and photochemical reactions [13,14,15]. To achieve the switching, a molecular design that enables activation/deactivation of the photosensitization under the desired conditions is required.

We have previously shown that a gold nanoparticle (AuNP) efficiently quenches the excited state of porphyrin when they are covalently linked to form a conjugate [16,17,18]. In the porphyrin–AuNP conjugates, [16] ^1^O_2_ generation was almost quenched (singlet oxygen generation quantum yield, Φ_Δ_ = 0.01−0.08) from corresponding unbound porphyrin (Φ_Δ_ = 0.70 for *meso*-tetraphenyl porphyrin (TPP) [19], at a partial pressure of oxygen *p*O_2_ = 0.21 atm). Recently, we have established a switchable photosensitization system for ^1^O_2_ generation driven by an acid–base reaction that incorporates a supramolecular architecture [18]. This switch is based on a reversible change of the distance between the porphyrin and the AuNP, a so-called shuttling motion [20]. The porphyrin is topologically interlocked with the AuNP shuttles between two ‘stations’ placed on an axle molecule, which is introduced perpendicular to the AuNP surface via a S–Au covalent linkage. This supramolecular switch has an advantage in that it is possible to design a switch that responds to a desired stimulus through the molecular design of the supramolecular part, and reversible switching is also possible. However, the on/off ratio of Φ_Δ_ was only 1.9 (Φ_Δon_/Φ_Δoff_ = 0.097/0.052), suggesting that the excitation energy transfer is still dominant in the on-state (where the porphyrin is located at the station away from the AuNP).

Here, we demonstrate a method to switch the generation of ^1^O_2_ by sulfide ligand exchange reactions on the porphyrin–AuNP conjugates. The ligand exchange reaction between the conjugates with sulfide compounds (thiol or disulfide) resulted in the porphyrin desorption from the AuNP quencher surface and restored Φ_Δ_. The on–off ratio depends on the type of incoming sulfide, suggesting that the exchange reaction is thermodynamically or kinetically controlled. Even after complete porphyrin desorption, light absorption by AuNPs in the system still suppressed the ^1^O_2_ generation. With proper molecular design of the incoming sulfide, the AuNPs can be precipitated simultaneously with the porphyrin desorption to restore Φ_Δ_.

## 2. Results and Discussion

### 2.1. Synthesis of Porphyrin–AuNP Conjugates Via Disulfide/Thiolate Ligand Exchange Reaction

Common methods for introducing desired functional groups onto the surface of thiolate-protected AuNPs can be classified into two approaches [21]: (i) a direct synthesis method in which thiols and gold ions (typically Au^+^ or Au^3+^) are allowed to coexist for reduction, and (ii) a post-synthetic method in which separately synthesized thiolate-protected AuNPs are mixed with a functional group-appended sulfide compound to allow for a sulfide ligand exchange reaction. Figure 1 shows a schematic diagram of the sulfide ligand exchange reaction [22]. In the reaction between thiol (R^1^SH) and thiolate-protected AuNP (R^2^S–AuNP, only one ligand is shown for simplicity), the thiol is introduced onto the AuNP surface via a proton exchange reaction with the thiolate ligand (R^2^S) on the AuNP surface (thiol–thiolate (T–T) exchange reaction, Figure 1A). A similar exchange reaction occurs when the disulfide (R^1^SSR^1^) is used instead of the thiol (disulfide–thiolate (D–T) exchange reaction, Figure 1B). The D–T exchange reaction is much slower than the T–T exchange reaction, so it is a method that is not often used, but it has the advantage of being easy to operate due to the stability of disulfides against oxidation [23]. The disulfide R^1^SSR^1^ reacts nucleophilically with the thiolate ligand R^2^S on the AuNP surface to give the asymmetric disulfide R^1^SSR^2^. Thus, the atom economy of this reaction step is 0.5, but the remaining asymmetric disulfide can again react with the thiolate on the AuNP surface to achieve a thermodynamically determined equilibrium. Since the T–T exchange reaction is fast, it usually proceeds at room temperature, while the D–T reaction often requires heating. A 1:1 stoichiometry is assumed here for the incoming sulfide and the outgoing sulfide [24], which is known to not always be the case but will not be discussed here.

Figure 2 shows the chemical structure of the present porphyrin–AuNP conjugate. We designed a disulfide-type porphyrin ligand (**1**) to introduce porphyrin onto the surface of AuNPs via a D–T exchange reaction. This compound is stable to air and moisture, so it can be handled without special precautions, unlike thiols. The AuNPs, into which porphyrin is going to be introduced, were synthesized by the two-phase method reported by Brust et al. [25], where a gold ion (Au^3+^) is reduced in the presence of 1-dodecanethiol (2). The resulting 1-dodecanethiolate-protected AuNPs (**2**@AuNP) are highly soluble in common organic solvents which allow ligand exchange reactions to proceed in homogeneous solutions. The nanoparticle diameter could be predicted to be ca. 2.5 nm because the AuNPs with an average composition of Au_400_(C_12_H_25_S)_126_ were synthesized under identical conditions [26].

When **1** was mixed with **2**@AuNP in toluene (*ca.* 6.15:1 in mole, see details in experimental section), the D–T exchange reaction was not observed at room temperature. On the other hand, a higher reaction temperature (at 60 °C and 80 °C) allowed for the introduction of porphyrins onto the AuNP surface by promoting the exchange reactions. Figure 1A,B show the UV–vis absorption spectra of the porphyrin–AuNP conjugates (**1**@AuNP) after purification by reprecipitation. Judging from the characteristic Soret band (around 420 nm), the exchange reaction was not complete even after 24 h at 60 °C, while the reaction reached equilibrium in about 10 h at 80 °C (Figure 1C). These D–T exchange reactions were well approximated as pseudo-first-order reactions, and the rate constants were *k*_60°C_ = 1.1 × 10^−5^ and *k*_80°C_ = 9.9 × 10^−5^ [s^−1^], respectively. These values are smaller than known D–T exchange reactions, possibly due to the bulkiness of the porphyrins and the short linkers [27,28].

In the purification process by reprecipitation, the supernatant after centrifugation contained almost no porphyrin after 10 h when the D–T exchange reaction was performed at 80 °C (judging from the characteristic violet color of porphyrin), indicating that the introduction of **1** is thermodynamically favorable. This seems surprising given the highly sterically hindered structure of **1**, but may be influenced by the thermodynamic stability of the remaining disulfide (i.e., didodecyl disulfide). Assuming that all porphyrins have been introduced, approximately 12.3 porphyrin ligands have been introduced per AuNP. In subsequent experiments, **1**@AuNP which reacted at 80 °C for 24 h was used.

### 2.2. Ligand Exchange Reaction with 1-Dodecanethiol

#### 2.2.1. Desorption Monitoring by UV–Vis Absorption Spectra

To investigate the applicability of the sulfide ligand exchange reaction to the switch in ^1^O_2_ generation, we preliminarily investigated whether porphyrin desorption is possible using the T–T exchange reaction between **1**@AuNP and **2**. In the UV–vis absorption spectrum, a slight broadening of the Soret band is observed when porphyrin is introduced onto the AuNP surface. Similar broadening is also widely observed in other dye–AuNP conjugates and is mainly attributed to the exciton coupling and restricted molecular motion of the chromophore [29,30,31]. Using this property, the desorption of the porphyrin by the ligand exchange reaction can be monitored by the absorption spectra. When a large excess **2** was added to **1**@AuNP in toluene at room temperature, the Soret band (the absorption maxima = 419 nm) sharpened with time, suggesting the desorption of the porphyrin (Figure 2). The full width at half maximum was asymptotic from 13.1 nm to 12.0 nm, in which the latter is equivalent to that of TPP [18], in about 10 h. This observation suggests that almost all porphyrin ligands were desorbed from the AuNP surface.

#### 2.2.2. Concentration Dependence on Singlet Oxygen Generation Quantum Yield (Φ_Δ_)

The quantum yield of the ^1^O_2_ generation (Φ_Δ_) after the T–T exchange reaction with **2** at different stoichiometries ([**2**] = 0–50 mmol/L, up to 2500 equivalent) was determined by the chemical quencher method (details are given in the experimental section). Φ_Δ_ can be obtained by monitoring the decomposition of 1,3-diphenylisobenzofuran (DPBF) by light irradiation in the presence of photosensitizer [19]. A long-pass filter with a cut-on wavelength of 500 nm was inserted into the optical path to avoid direct photodegradation of DPBF. Therefore, the photoexcitation of the porphyrins occurred only in the Q-band (500–700 nm, corresponding to the S_0_→S_1_ transition), and the Soret band (around 420 nm, S_0_→S_2_ transition) did not contribute to the generation of ^1^O_2_.

The Φ_Δ_ of the **1**@AuNP before ligand exchange was determined to be 0.08, which is close to the results for our previously reported porphyrin–AuNP conjugates [16]. In contrast, a clear restoration of Φ_Δ_ was observed after the ligand exchange reaction with **2** (Figure 3). This is because the porphyrin was desorbed from the AuNP surface by the ligand exchange reaction, and the excitation energy transfer from porphyrin to the AuNP was eliminated. Under the conditions investigated ([**1**@AuNP] = 0.20 μmol/L, room temperature, 10 h), Φ_Δ_ was asymptotic to about 0.4 at the concentration of **2**, which was about 10 mmol/L, suggesting that the porphyrin ligands from the AuNP surfaces were completely desorbed above this concentration. However, this Φ_Δ_ value is still only about half that of the TPP (Φ_Δ_ = 0.70). This may be due to the following factors. As apparent from the absorption spectrum, in the irradiation wavelength region (>500 nm), the irradiated light is absorbed by the AuNPs rather than the porphyrins. For example, the absorbance of AuNPs at 515 nm at this concentration is approximately 0.2, which corresponds to a transmittance of 63%. In contrast, porphyrins have an absorbance of only about 0.05. This is where the porphyrin contribution is greatest (highest Q-band molar absorption coefficient), so a significant fraction of the light is absorbed by the AuNPs over the entire wavelength range. Another possibility is that there is excitation energy transfer from the unbound porphyrins to the AuNPs, but this would make a small contribution under such dilute conditions.

### 2.3. Ligand Exchange Reaction with Various Sulfide Compounds

To confirm that the recovery of Φ_Δ_ is due to the ligand exchange reaction, we performed similar experiments with various sulfide compounds **3**–**12** (Figure 4). The concentration of the sulfide compound was set to 1.0 mmol/L to clarify the differences between the compounds, where the ligand exchange reaction was not completed in the case of **2** (other conditions are fixed: [**1**@AuNP] = 0.20 μmol/L, room temperature, 10 h) (Table 1). The Φ_Δ_ of the ligand exchange reaction product with **2** was 0.24 ± 0.01, while that of the corresponding disulfide **3** was 0.05 ± 0.03, almost unchanged from **1**@AuNP before the exchange reaction, indicating that no D–T exchange reaction occurred (D–T exchange reactions are much slower than corresponding T–T exchange reactions) [23]. This trend was similar for thiophenol (**4**) and the corresponding disulfide (**5**) (Φ_Δ_ after ligand exchange reaction: 0.24 ± 0.01 (**4**); 0.14 ± 0.02 (**5**)).

In the ligand exchange reaction product with compounds **2**–**5**, there was no significant change in the absorption spectra, except for the above-mentioned sharpening of the Soret band, suggesting that the AuNPs remain stably dispersed in the solution. On the other hand, in the absorption spectrum of the ligand exchange reaction product with 3-mercaptopropionic acid (**6**), the absorption band of the AuNPs was reduced indicating that the aggregation of the AuNPs took place (Figure 5). Black precipitates were formed and Φ_Δ_ was restored to 0.42 ± 0.09, indicating that the light absorption by the remaining AuNPs was somewhat eliminated. The absorbance from the porphyrin did not decrease, indicating that the porphyrin ligands were not precipitated together with the AuNPs. Consequently, by introducing a polar functional group (i.e., carboxyl group, –COOH) to the terminal of the incoming thiol, the simultaneous porphyrin desorption and the precipitation removal of the AuNPs can be achieved (Figure 3A). The disulfide (**7**) corresponding to **6** was not soluble in toluene, so its phenyl ester (**8**) was used. Similar to the other disulfides (**3** and **5**), no restoration of Φ_Δ_ was observed (Φ_Δ_ = 0.07 ± 0.01).

More prominent AuNP precipitation was observed in a ligand exchange reaction with a divalent thiol, dihydrolipoic acid (also called dihydrothioctic acid) (**9**) (Figure 5), where Φ_Δ_ was restored to 0.59 ± 0.02 comparable to that of TPP. This is likely due to a more efficient ligand exchange reaction compared to the monovalent **6**. In fact, when the carboxyl terminal was masked by an *n*-butyl group (i.e., compound **10**), the ligand exchange product showed a Φ_Δ_ of only 0.30 ± 0.01, which is slightly larger than the other non-polar thiols (i.e., **2** and **4**). The crosslinking between the AuNPs by divalent thiols was also thought to be the cause of the precipitation (Figure 3B) [32]. However, aggregation was not observed, probably because the distance (three carbon atoms) intramolecularly between the two thiols is insufficient to form the crosslinking.

The case of the intramolecular disulfide (dithiolane) corresponding to **9** (compound **11**) is a little more complicated. Similar to other disulfide compounds (**3**, **5**, and **8**), no significant restoration of Φ_Δ_ was observed (Φ_Δ_ = 0.10 ± 0.03). However, slight precipitation of the AuNPs was found in the ligand exchange reaction product (Figure 5), suggesting that the D–T exchange reaction occurs at room temperature, and a small amount of introduced carboxylic groups promote the aggregation. Based on the mechanism of the D–T exchange reaction in Figure 1, the porphyrin ligand can be retained as a disulfide on the AuNP surface via simultaneous or successive exchange reaction(s) (Figure 3C).

From the above results, the only structural requirement of the incoming ligand for efficient Φ_Δ_ restoration is the thiol group (–SH). The cysteine derivative **12** could also be involved in the T–T exchange reaction. However, the Φ_Δ_ after the exchange reaction was 0.15 ± 0.01, which was the smallest restoration among thiols examined in this study, indicating that the ligand exchange reaction does not proceed efficiently due to bulkiness around the thiol group [28].

## 3. Materials and Methods

All chemicals were obtained from commercial sources and used without purification unless otherwise noted. Hydrogen chloride solution in *n*-butanol was prepared by passing dry hydrogen chloride through *n*-butanol. 1,3-Diphenylisobenzofuran (DPBF) was purchased from Sigma-Aldrich (St. Louis, MO, USA). For the spectra measurement and singlet oxygen generation experiment, freshly distilled and air-saturated toluene was used. The concentration of oxygen in the air-saturated toluene is 1.93 mM at atmospheric pressure (*p*O_2_ = 0.21 atm) [33].

### 3.1. Synthesis

**1**: To a mixture of 5-(4-hydroxyphenyl)-10,15,20-triphenylporphyrin (315 mg, 0.50 mmol), dry pyridine (1 mL) and dry chloroform (100 mL), 3,3′-dithiodipropionyl dichloride (173 mg, 0.70 mmol) in dry chloroform (1.4 mL) was gradually added. After 20 h of stirring, the mixture was washed with water, dried over sodium sulfate and concentrated in vacuo. The residue was purified by column chromatography (Wakogel C-200, methylene chloride) and subsequent reprecipitation was carried out from toluene/hexane to obtain **1** as a purple powder (0.170 g, 47%). CAS No. 2093202-53-8. ^1^H NMR (400 MHz, chloroform-*d*, SiMe_4_, RT): δ/ppm −2.80 (brs, 4H, NH), 3.29–3.33 (m, 8H, CH_2_), 7.56 (m, 4H, Ar), 7.65–7.80 (m, 18H, Ar), 8.13–8.26 (m, 16H, Ar), 8.78–8.26 (m, 16H, pyrrole).

**1-Dodecanethiolate-protected AuNP (2@AuNP):** synthesized from tetrachloroauric acid and **2** according to the literature [25].

**1@AuNP:** Two hundred microliters each toluene solution of 2@AuNP (50 mg/mL = 0.48 mmol/L, when assuming chemical composition of Au_400_(C_12_H_25_S)_126_ = 1.04 × 10^5^ kg/mol) and **1** (4.23 mg/mL = 2.95 mmol/L) were placed in a glass vial together with a stir bar, and heated and stirred in an oil bath at 60 °C or 80 °C. At the certain time of reaction, an aliquot (10 μL) was transferred into a 1.5 mL centrifugal tube to which acetone (1 mL) was added, and then the contents were mixed rapidly by finger tapping. The conjugates were precipitated by centrifugation (10,000× *g*, 1 min) and the supernatant was discarded. After washing the precipitate by repeated redispersing and sonicating it in acetone, the precipitate was finally dried under reduced pressure.

**Didodecyl disulfide (3):** To a solution of 1-dodecanethiol (**2**) (1.0 g, 4.9 mmol) in ethanol (25 mL), elemental iodine crystals were slowly added until the color of the solution turned to pale yellow. The resulting white crystals were collected by suction filtration and the filtrate was concentrated to around half the volume. The second crystals were collected, and the combined crystals were recrystallized from chloroform/methanol to afford **3** as white crystals (0.872 g, 88%). CAS No. 2757-37-1. ^1^H NMR (400 MHz, chloroform-*d*, SiMe_4_, RT): δ/ppm 0.88 (t, 6H, CH_3_), 1.20–1.45 (m, 36H, CH_2_), 1.67 (quintet, 4H, 2-CH_2_), 2.68 (t, 4H, 1-CH_2_).

**Diphenyl 3,3′-dithiodipropionate (8):** To a mixture of phenol (0.94 g, 10 mmol) and dry pyridine (2.9 mL, 36 mmol) in dry chloroform (40 mL), a solution of 3,3′-dithiodipropionyl dichloride (2.0 g, 8 mmol) in dry chloroform (40 mL) was added over 2 h. The mixture was warmed up to room temperature and stirred for another 1 h. After the removal of the solvent, the residue was purified by column chromatography (Wakogel C-200, hexane:methylene chloride = 1:1) to afford **8** as a yellow solid (1.27 g, 70%). CAS No. 2376483-56-4. ^1^H NMR (400 MHz, chloroform-*d*, SiMe_4_, RT): δ/ppm 2.95–3.10 (m, 8H, CH_2_), 7.09 (m, 4H, ortho-H), 7.22 (m, 2H, para-H), 7.36 (m, 4H, meta-H).

**Dihydrolipoic acid (9):** A mixture of DL-lipoic acid (**11**) (0.525 g, 2.5 mmol) and sodium bicarbonate (0.210 g, 2.5 mmol) in water (12.5 mL) was sonicated until all the substances were completely dissolved. The mixture was cooled in an ice bath; sodium borohydride (0.475 g, 12.5 mmol) was added dropwise in 20 min intervals and stirred for 30 min. After additional stirring for 30 min, the mixture was acidified to pH 1 by adding 2 M HCl under nitrogen bubbling. The mixture was extracted by chloroform (3 × 10 mL), dried over magnesium sulfate, and then concentrated in vacuo to afford **9** as a colorless oil (0.427 g, 82%). CAS No. 462-20-4. ^1^H NMR (400 MHz, chloroform-*d*, SiMe_4_, RT): δ/ppm 1.31 (d, 1H, 6-SH), 1.36 (t, 1H, 8-SH), 1.40-1.96 (m, 8H, 3,4,5,7-CH_2_), 2.38 (t, 2H, 2-CH_2_), 2.62–2.80 (2H, 8-CH_2_), 2.93 (m, 1H, 6-CH), 10.56 (brs, 1H, COOH).

***n*-Butyl dihydrolipoate (10):** Compound **9** (1.91 g, 9.2 mmol) was dissolved in hydrogen chloride (ca. 10% in *n*-butanol, 15 mL). After 3 h of stirring at room temperature under nitrogen, the mixture was concentrated in vacuo. The residue was dissolved in toluene, and concentrated in vacuo to remove *n*-butanol as an azeotrope, which was repeated three times. The residue was redissolved in methylene chloride, washed with water (3×), dried over sodium sulfate, and concentrated in vacuo to afford **10** as a colorless liquid (2.39 g, 99%). CAS No. 245112-97-4. ^1^H NMR (400 MHz, chloroform-*d*, SiMe_4_, RT): *δ*/ppm 0.94 (t, 3H, butyl CH_3_), 1.30 (d, 1H, 6-SH), 1.35 (t, 1H, 8-SH), 1.38 (m, 2H, butyl-CH_2_), 1.40–1.70 (m, 9H), 1.70–1.96 (m, 2H, 7-CH_2_), 2.32 (t, 2H, 2-CH_2_), 2.60-2.80 (m, 2H, 8-CH_2_), 2.92 (m, 1H, 6-CH_2_), 4.07 (t, 2H, butyl-CH_2_).

***N*-Acetyl-L-cysteine *n*-butyl ester (12):** This was synthesized in a similar manner as **10** from *N*-acetyl-L-cysteine (1.63 g, 10 mmol). The compound was finally reprecipitated from diethyl ether/hexane, washed with cold hexane, and dried to obtain **12** as a white solid (1.24 g, 62%). CAS No. 34233-59-5. ^1^H NMR (400 MHz, chloroform-*d*, SiMe_4_, RT): *δ*/ppm 0.95 (t, 3H, CH_3_), 1.33 (t, 1H, SH), 1.40 (sextet, 2H, butyl-CH_2_), 1.66 (quintet, 2H, butyl-CH_2_), 2.08 (s, 3H, COCH_3_), 3.03 (dd, 2H, cysteine-CH_2_), 4.04–4.27 (m, 2H, butyl-CH_2_), 4.88 (m, 1H, systeine-CH), 6.41 (brs, 1H, NH).

### 3.2. Sulfide Ligand Exchange Reaction

The toluene solution of **1**@AuNP (25 μg/mL) and sulfide compound **2**–**12** were mixed at a 4:1 volume ratio and stirred at room temperature to allow for the ligand exchange reaction. The mixture was used for subsequent characterization without further treatment.

### 3.3. Determination of Singlet Oxygen Quantum Yield (Φ_Δ_)

All Φ_Δ_ measurements were obtained in a dark room at a maintained temperature (25 °C). To the sample solution, 1,3-diphenylisobenzofuran (DPBF) (1.2 mmol/L in toluene) was added to make its concentration 20 μmol/L. The sample was filled in a quartz cuvette (1 cm × 1 cm) and irradiated by visible light (500 W Xe short arc lamp, USHIO optical modulex. UV-blue light (<500 nm) from the light source was cut off by passing it through an optical filter Y-50 to avoid the undesired decomposition of DPBF). The light irradiation period (typically 3 s) was controlled by a shutter and the decay of the DPBF absorption at 416 nm was recorded by a UV–vis spectrometer. The irradiation cycles were repeated until ~10% of DPBF was decomposed. This procedure was repeated at least four times for each sample. *meso*-Tetraphenylporphyrin (TPP) was used as the reference compound (1.00 × 10^−7^ M in toluene, Φ_Δ_ = 0.70 at *p*O_2_ = 0.21 atm) and the decay rate was measured by the same method used for the conjugates.

The singlet oxygen quantum yields (Φ_Δ_) were determined by the procedure previously reported [16,18,19]. First, to eliminate the effect of light absorption by AuNPs, the molar extinction coefficient arising from porphyrin ligands was deconvoluted from whole conjugate spectra using the following equation:$$A\left(\lambda \right)=a\times \varepsilon_{\mathrm{Por}}\left(\lambda \right)+b\times \varepsilon_{\mathrm{AuNP}}\left(\lambda \right)+R\left(\lambda \right)$$

Here, *A*(*λ*) is the actual extinction spectrum of **1**@AuNP. *ε*_Por_(*λ*) and *ε*_AuNP_(*λ*) are the separately measured molar absorption coefficients of TPP and **2**@AuNP, respectively. The proportional coefficients *a* and *b* were determined by the non-linear least squares method (300 ≤ λ ≤ 800 nm), and the residue *R*(*λ*) was obtained. Combining the above results, Φ_Δ_ were calculated by the following equation
$$\Phi_{\Delta }=\Phi_{\Delta \mathrm{ref}}\times \frac{A_{\mathrm{ref}}}{A_{}}\times \frac{r_{}}{r_{\mathrm{ref}}}$$
where *A* is the area of porphyrin absorption spectra, *r* is the decomposition rate of DPBF measured by a UV–vis spectrometer, and subscript R refers to a standard with known Φ_Δ_ (i.e., TPP, Φ_Δ_ = 0.70), respectively.

## 4. Conclusions

By incorporating porphyrin ligands onto the AuNP surface via a covalent linkage, ^1^O_2_ generation was almost quenched due to efficient excited energy transfer from the porphyrin to the AuNP. The porphyrin ligands were desorbed from the surface of the AuNPs by the reverse ligand exchange reaction and showed restored ^1^O_2_ generation ability with Φ_Δon_/Φ_Δoff_ of up to 7.4 (0.59/0.08, incoming sulfide = dihydrolipoic acid 9), which was significantly different depending on the type of sulfur compound used. This difference was influenced by the efficiency of the ligand exchange reaction, as well as the aggregation of the AuNPs governed by the polarity of the protecting ligand layers. These results indicate that the adsorption/desorption between photosensitizers and AuNPs can be significantly applicable to the switching of ^1^O_2_ generation.

## Data Availability

The raw data supporting the conclusions of this article will be made available by the authors, without undue reservation.

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
