# Peer review of "Singlet Oxygen Generation Driven by Sulfide Ligand Exchange on Porphyrin–Gold Nanoparticle Conjugates"

_ijms, 2023, doi:10.3390/ijms24087600_

Round 1
Reviewer 1 Report
The authors have described a method to control singlet oxygen generation through sulfide ligand exchange reactions in gold-porphyrin nanoparticle (AuNP) conjugates. The method involves adsorption and desorption of porphyrins on the surface of gold nanoparticles driven by sulfide compounds. The on/off ratio of the singlet oxygen quantum yield can be controlled both thermodynamically and kinetically, depending on the type of incoming sulfide. The subject is interesting, and their research is commendable. However, it would be beneficial to include a more comprehensive characterization of the gold nanoparticles using techniques such as TEM or STEM.
Author Response
Thanks for the constructive suggestions, we agree that TEM and STEM would be beneficial for the evaluation of present nanomaterials. However, sincerely sorry, we do not have good access to them at the moment. We will take this into consideration when we develop this system in the future. Therefore, we cited literature values.(line98)
Reviewer 2 Report
The manuscript addresses a novel method of switching 1O2 generation by the adsorption/desorption between photosensitizers and AuNPs. The overall experiment design and data analysis are sound. However, some minor issues need to be addressed.
Line 14:
was reached to 7.4 à reached 7.4
Throughout examining à By examining
Line 16:
The gold nanoparticles remained in the system à The remaining gold nanoparticles in the system
Line 52-53:
It is not clear why the dependence of on-off ratio on the type of incoming sulfide suggests kinetically and thermodynamically controlled exchange reaction. Author should consider either removing the latter half of the statement, or adding more supporting evidence.
Line 92:
So can be handled à so it can be handled
Line 239:
Consider either move section 3 right after the introduction, or after the conclusion.
In addition, the manuscript need further language editing to make it more concise.
Author Response
>>It is not clear why the dependence of on-off ratio on the type of incoming sulfide suggests kinetically and thermodynamically controlled exchange reaction. Author should consider either removing the latter half of the statement, or adding more supporting evidence.
So far, it is not certain whether the exchange reaction that determines the on-off ratio has reached equilibrium at the time of interest. We have revised the related description to make consistence with the results.
>>Consider either move section 3 right after the introduction, or after the conclusion.
This suggestion will make this paper more readable. However, the IJMS Instructions for Authors state that the experimental section should be placed before the conclusion. For this reason, the original order will be retained.
>>some minor issues need to be addressed.
Minor typos and other minor typos have been thoroughly revised, including those pointed out by the reviewer.
Reviewer 3 Report
In this study, authors have analyzed a switching method for singlet oxygen generation using on adsorption/desorption of porphyrins on AuNPs promoted by sulfide chemistry (gold-thiol bond). The study carries high significance and could be considered for publication. I have some suggestions to author prior to its final recommendation.
Some Comments:
In Scheme 1 and 2, it would be appropriate to mention the size of nanoparticles and depict the number of conjugates attached per particle.
Figure 1, a bar graph or line graph with respect to time points could be better to visualize the normalized abs at given wavelength.
Similar to above comment, in Figure 5, additional bar for the highest peak observed can me drawn to show abs peak at a particular wavelength.
In conclusions, provide a key significant results and statistical values along with future implications of the study.
Author Response
>>In Scheme 1 and 2, it would be appropriate to mention the size of nanoparticles and depict the number of conjugates attached per particle.
We added the relevant description in the Scheme 2 caption.
>>Figure 1, a bar graph or line graph with respect to time points could be better to visualize the normalized abs at given wavelength.
Figure 1C and 2B were changed to line graph (line graph is not applicable to Fig 3 due to lack of connection with each points). We believe this modification improve readability of the manuscript.
>>Similar to above comment, in Figure 5, additional bar for the highest peak observed can me drawn to show abs peak at a particular wavelength.
Absorbance at 450 nm for each sample were added as panel B to visualize the aggregation f AuNP.
>>In conclusions, provide a key significant results and statistical values along with future implications of the study.
Thank you for the constructive suggestion. Conclusion part was revised accordingly.